# Analysis of policy interventions to attract and retain nurse midwives in rural areas of Malawi: A discrete choice experiment

**Leslie Berman**[1]*, **Levison Nkhoma**[1], **Margaret Prust**[2], **Courtney McKay**[2], **Mihereteab Teshome**[1], **Dumisani Banda**[3], **Dalitso Kabambe**[4], **Andrews Gunda**[1]

1 Clinton Health Access Initiative, Inc. (CHAI) Malawi, Lilongwe, Malawi, 2 Analytics and Implementation Research Team, Clinton Health Access Initiative, Inc. (CHAI), Boston, Massachusetts, United States of America, 3 Department of Human Resource Management and Development, Ministry of Health and Population, Lilongwe, Malawi, 4 Department of Planning and Policy Development, Ministry of Health and Population, Lilongwe, Malawi

* leslie.r.berman@gmail.com

## Abstract

### Background

Inadequate and unequal distribution of health workers are significant barriers to provision of health services in Malawi, and challenges retaining health workers in rural areas have limited scale-up initiatives. This study therefore aims to estimate cost-effectiveness of monetary and non-monetary strategies in attracting and retaining nurse midwife technicians (NMTs) to rural areas of Malawi.

### Methods

The study uses a discrete choice experiment (DCE) methodology to investigate importance of job characteristics, probability of uptake, and intervention costs. Interviews and focus groups were conducted with NMTs and students to identify recruitment and retention motivating factors. Through policymaker consultations, qualitative findings were used to identify job attributes for the DCE questionnaire, administered to 472 respondents. A conditional logit regression model was developed to produce probability of choosing a job with different attributes and an uptake rate was calculated to estimate the percentage of health workers that would prefer jobs with specific intervention packages. Attributes were costed per health worker year.

### Results

Qualitative results highlighted housing, facility quality, management, and workload as important factors in job selection. Respondents were 2.04 times as likely to choose a rural job if superior housing was provided compared to no housing (CI 1.71–2.44, p<0.01), and 1.70 times as likely to choose a rural job with advanced facility quality (CI 1.47–1.96, p<0.01). At base level 43.9% of respondents would choose a rural job. This increased to 61.5% if superior housing was provided, and 72.5% if all facility-level improvements were provided,

**Data Availability Statement:** All relevant data are within the paper and its Supporting Information files.

**Funding:** This study was made possible with financial support from the Norwegian Ministry of Foreign Affairs through a grant from the Royal Norwegian Embassy of Malawi (https://www.norway.no/en/malawi/) to the Clinton Health Access Initiative Malawi. The funders had no role in study design, data collection and analysis, decision to publish, or preparation of the manuscript. The lead author and five of the co-authors are employees of the Clinton Health Access Initiative. Through the grant from the Royal Norwegian Embassy of Malawi to the Clinton Health Access Initiative (CHAI), CHAI provided support for study implementation and for the authors, in the form of salary support. These authors, as study investigators, supported the study design, data collection and analysis, and preparation of this manuscript together with the other coauthors who are employees of the Ministry of Health of Malawi.

**Competing interests:** Six authors are employees of the Clinton Health Access Initiative, including the lead author. The Clinton Health Access Initiative (CHAI) funded this study through a grant from the Royal Norwegian Embassy of Malawi to CHAI. This affiliation does not alter our adherence to PLOS ONE policies on sharing data and materials.

compared to an urban job without these improvements. Facility-level interventions had the lowest cost per health worker year.

## Conclusions

Our results indicate housing and facility-level improvements have the greatest impact on rural job choice, while also creating longer-term improvements to health workers' living and working environments. These results provide practical evidence for policymakers to support development of workforce recruitment and retention strategies.

## Introduction

Inadequate and unequal distribution of health workers are significant barriers to the provision of essential health services in Malawi. Malawi faces health workforce shortages of 48% against its national targets, with only 1.48 health workers per 1,000 population [1], far below the WHO recommended minimum density of 4.45 doctors, nurses and midwives per 1,000 population for countries to meet the Sustainable Development Goals [2]. Workforce shortages are particularly acute in rural areas where 84% of Malawi's population resides, contributing to disparities in access to health services and health outcomes between urban and rural areas [3,4]. For example, in rural areas in 2014 there were 0.7 clinicians per 1,000 persons as compared to 1.8 per 1,000 persons in urban areas [5].

In 2004, to respond to severe health workforce shortages, the Government of Malawi began implementing a 6-year Emergency Human Resources Programme (EHRP). The EHRP increased the number of health workers in 11 priority cadres from 5,453 to 8,369 by 2009, achieving a provider-to-population ratio of 1.44 per 1,000 population [6]. Among prioritized cadres, the EHRP supported the rapid scale up of Nurse Midwife Technicians (NMTs), a 3-year diploma nursing cadre drawn predominantly from rural areas and meant to serve in rural health facilities. NMTs are core to Malawi's primary care architecture, with the largest number of established posts among facility-based cadres [1]. Despite large investments during the EHRP period, which led to a 39% increase in the number of nurses to 4,812 by 2009, there have been minimal gains over the past decade [6]. In 2017, there were 5,441 nurses and a vacancy rate of 62% against established posts [1]. Retention of NMTs and other critical cadres has remained a major challenge and has limited the effectiveness of workforce scale-up initiatives. National attrition estimates range from 3% to 15%, but likely underestimate actual attrition as these data are not routinely captured [1,7].

Malawi's Human Resources for Health (HRH) Strategic Plan 2017–2022 emphasizes the importance of improving retention and motivation of health workers as critical to effective, efficient and equitable health service delivery. Qualitative literature has pointed to key strategies for rural recruitment and retention in low- and middle-income countries (LMICs), including educational and development opportunities, financial incentives, improved living and working conditions, staff recognition and management improvements, and regulatory policy changes such as task-shifting, creation of new cadres, and compulsory service agreements [8,9].

Job choices are highly context specific and the effectiveness of different strategies in facilitating a decision to take a job can vary greatly across countries and cadres, with the most effective strategies responding to clear challenges in the national workforce landscape [10,11]. Health workforce research in Malawi has found health workers were unsatisfied with their

salaries, benefits, living conditions, workload, lack of supplies and management relationships [12–14]. A discrete choice experiment (DCE) conducted in Malawi in 2008 highlighted that nursing officers were willing to trade between various monetary and non-monetary benefits and that multiple attributes could have a significant impact on recruitment; however, this study did not investigate the differential effects of these incentives in rural and urban areas, the ability of the incentives to attract health workers to rural areas, or the costs of incentives relative to their expected impact [15]. While previous evidence aimed to understand drivers of health worker retention in general, an investigation of factors that can influence rural job uptake and retention was needed, with a focus on mid-level providers who are the frontline workers in rural facilities in Malawi. We conducted a DCE to estimate the cost-effectiveness of policy-relevant monetary and non-monetary strategies in attracting and retaining public sector NMTs to job posts in rural areas of Malawi.

## Methods

This study used a DCE methodology to capture information on the relative importance of different job characteristics, the probability of uptake of jobs defined by those characteristics and associated costs of the interventions. The study was conducted in two phases, including a) interviews, focus group discussions, and government consultations to identify incentives for inclusion in the DCE questionnaire and b) administration of the DCE questionnaire to practicing NMTs and NMT students. This study was approved by the Malawi National Health Sciences Research Committee (protocol approval number NHSRC #15/3/1394), and the U.S.-based Chesapeake Ethics Review Board (Pro00013609). Written informed consent was obtained from all study participants.

DCEs are a useful tool to provide quantitative information on the relative importance of various job characteristics that influence the job choices of health workers, as well as the trade-offs between these factors and thus the probability of uptake of jobs [16]. This method goes beyond more frequent qualitative assessments, and through asking participants to choose between different job scenarios, DCEs can be used to provide quantifiable data to guide the selection of the most appropriate strategies for recruitment and retention in underserved areas [16–19].

### Qualitative data collection and analysis

Interviews and focus group discussions were used to identify factors that would motivate NMTs and NMT students to select and remain in a rural job using a purposive sampling approach to reach a diverse group of urban and rural NMTs and students, with the aim of achieving data saturation. Three focus group discussions were conducted in three different schools, with each focus group including 12 NMT students in their final year of study who were randomly selected from the class register. At the time of the study, there were 12 schools that trained NMTs in Malawi, and one school was purposively selected from each of the three regions of the country (northern, central and southern). Twelve in-depth interviews were conducted with practicing NMTs serving in both rural and urban facilities operated by the Ministry of Health (MoH) and the Christian Health Association of Malawi (CHAM). CHAM is a large, faith-based non-governmental healthcare provider in Malawi, providing approximately 30% of Malawi's healthcare through service level agreements with the MoH. One district from each region was purposively selected and within each of the three selected districts four facilities were chosen, including one with each of the following classifications: MoH rural, MoH urban, CHAM rural and CHAM urban. Within those facilities, one NMT was selected who was available at the time of interview.

The discussion guide asked questions about participants' preferences on job characteristics and is available in S1 File. Interviews and focus group discussions were conducted by three data clerks fluent in both Chichewa and English, who received training from the study team on the protocol and data collection methods. The interviews and focus group discussions were audio recorded and transcribed verbatim in English.

Transcripts were independently coded by two members of the research team in Dedoose, a web-based qualitative analysis software. Using an inductive approach, the team identified and coded attributes mentioned in the interviews and focus group discussions that participants identified as informing their decision to choose and remain in rural and hard to reach facilities. The qualitative analysis generated a list of 31 coded attributes.

## DCE questionnaire development

The 31 attributes that emerged from the qualitative analysis were reviewed in a consultation workshop with government staff from the Departments of Nursing and Midwifery Services, Planning and Policy Development, and Human Resources Management and Development, as well as civil society organizations and development partners. Through a facilitated exercise, the workshop participants reviewed each attribute and associated qualitative quotations, and then prioritized attributes for inclusion in the DCE questionnaire using several inclusion criteria, such as the strength of qualitative preferences and the feasibility of implementing those incentives in Malawi. The workshop goal was to ensure that all interventions included in the DCE questionnaire were policy relevant and could be practically implemented in Malawi. The selected attributes included: housing, facility quality, access to long-term career progression opportunities (upgrading), workload, supportive management, and choice of location. Salary considerations emerged during focus group discussions and salary was specifically included as an attribute to allow for more detailed cost comparisons in the analysis. Within each attribute category, a base level and one or more higher-level options were defined to represent incentives that the government may offer to attract and retain health workers. The attributes and levels are shown in Table 1.

A labeled design was used for this DCE, so that each job choice set included one rural and one urban job. The DCE attributes and levels were combined to create job choice sets using R 3.5.2 software. A fractional factorial design was used to select a fraction of the total job choice sets in a way that allows for estimation of preferences for all job profiles, not just those presented in the questionnaire [16]. R software was used to select choice sets that optimize D-efficiency, maximize level balance and orthogonality, and minimize overlap among attribute levels. The purpose of having an efficient design is to maximize the precision of estimated model parameters [20]. Twenty-four choice sets were created and randomly assigned to one of two questionnaire blocks (blocks A and B), and participants randomly received one of the two blocks. The aim of blocking was to reduce the burden on any individual respondent while still achieving optimal experimental design across all choice set options [20].

In the final DCE questionnaire, each participant was presented with a series of 12 choice sets that each described two potential employment scenarios. For each of the two scenarios, a description of each job attribute was provided based on the selection of one of the levels for that attribute and the job location. The 12 choice sets in block A of the DCE questionnaire are shown in S2 File. Study staff read consent to all respondents, explaining the attributes in detail, and elaborating that choices should be based on respondents' preferences of factors that would both motivate them to choose and remain in a particular job. In addition to the DCE choice sets, demographic and background questions were included in the questionnaire.

**Table 1. Job characteristics included in the discrete choice experiment questionnaire.**

| Attribute | Job attribute levels | Offered in rural jobs | Offered in urban jobs |
|---|---|:---:|:---:|
| **Housing** | **No housing or housing allowance provided** | ✓ | ✓ |
| | **Free basic housing provided** (eg. semi-detached house with two bedrooms) | ✓ | |
| | **Free superior housing provided** (eg. detached house with reliable electricity and three bedrooms) | ✓ | |
| **Facility quality** | **Basic** (e.g. unreliable electricity; equipment, drugs and supplies not always available) | ✓ | ✓ |
| | **Advanced** (e.g. reliable electricity; equipment, drugs and supplies always available) | ✓ | ✓ |
| **Access to long-term upgrading opportunities** | Eligible to apply for upgrading opportunities **after 4 years of service** | ✓ | ✓ |
| | Eligible to apply for upgrading opportunities **after 3 years of service** | ✓ | ✓ |
| **Workload** | **Heavy workload** (you work longer hours because the facility does not have enough staff) | ✓ | ✓ |
| | **Manageable workload** (you work within scheduled hours because the facility has sufficient staff) | ✓ | ✓ |
| **Supportive management** | The management at the facility is **not supportive and makes work more difficult** | ✓ | ✓ |
| | The management at the facility is **supportive and makes work easier** | ✓ | ✓ |
| **Salary** | **125,069 MWK per month** | ✓ | ✓ |
| | **156,365 MWK per month** (25% top-up) | ✓ | ✓ |
| | **187,604 MWK per month** (50% top-up) | ✓ | ✓ |
| **Choice of location** | You are **randomly assigned** to a health facility | ✓ | ✓ |
| | You are given a **choice of district** in which you will work | ✓ | ✓ |

## DCE sampling and data collection

The DCE instrument was pre-tested with five deployed NMTs and 40 second year NMT students. The full DCE questionnaire was anonymously self-administered by 472 participants in September and October 2016. The sample size was determined through review of literature which indicates that obtaining reliable estimates of preferences requires at least 30 to 50 respondents per sub-group to be analyzed [16]. The appropriate sample size is impacted by the number of attributes and levels, and the number of choice sets provided to each participant. The overall precision of DCE parameters is impacted by a balance between statistical efficiency and response efficiency [19]. Using multiple DCEs with simulated sample sizes to estimate the sample size where precision would improve, a DCE methodological review revealed that precision steadily increases when the sample size is below 150 and flattens at around 300 observations and above [20]. We therefore aimed to sample a minimum of 300 participants.

To reach the minimum target sample size, we used a census approach, and collected data from a total of 472 respondents. All graduating NMT students were invited to participate from five randomly selected training institutions of the 12 training institutions in-country, representing geographic diversity across the country's five administrative zones (north, central west, central east, southwest, southeast). Although there was overlap in schools selected for focus groups and for the DCE, individual students included in the focus groups were not eligible for the DCE. For practicing NMTs, ten districts were randomly selected, including two districts from each of the five zones. Districts selected in the first qualitative phase were excluded from the second phase. From each district, six rural facilities and one urban facility were randomly selected. Private facilities, health posts and village clinics were excluded from the sample. At each health facility, all NMTs present on the day of the survey were invited to participate. Data were entered electronically using EpiData with 100% double-entry to ensure accuracy. The full DCE dataset is available in S3 and S4 Files.

### DCE data analysis

Demographic, education, and work experience characteristics were analyzed using univariate, descriptive statistics. Bivariate logistic regression was then used to explore associations between participants' self-reported likelihood of working in a rural area in the future and various demographic or background characteristics. For the data from the DCE choice sets, a conditional logit model was developed to investigate the preferences for job attributes among respondents. The conditional logit model is based on three assumptions: (1) independence of irrelevant alternatives; (2) error terms are independent and identically distributed across observations; and (3) no preference heterogeneity across respondents. Goodness-of-fit criteria, including Akaike and Bayesian information criteria and pseudo $R^2$, were used to assess model fit. Dummy variables were established for each attribute level in a rural or an urban setting and the probability of choosing a job with a higher-level attribute compared to the base level was produced. To investigate potential impact of demographic characteristics on job attributes, we also ran separate conditional logit models for the various demographic sub-groups of the population.

An uptake rate or preference impact measure was calculated to estimate the percentage of health workers that would prefer a job posting that offers a specific package of strategies as compared to other job postings [21]. Several validity tests were conducted to determine the appropriateness of model specifications. Specifically, we investigated dominance and internal or predictive validity. Dominance indicates that a participant always selected job options on the basis of one attribute (such as always choosing the higher salary). Such behavior is in violation of the basic assumptions of random utility theory, which informs the DCE model design, and assumes that individuals make trade-offs between various characteristics when making choices [22,23]. Therefore, we examined the number of participants that always chose jobs that offered the highest level of any characteristic and excluded from our analysis the 79 respondents who expressed a dominant preference. To assess internal or predictive validity, we compared the percentage of participants that chose a job option to the uptake predicted by the model [23,24]. All analyses were performed using Stata 13.

## Results

### Qualitative findings on factors motivating rural uptake and retention

In the qualitative interviews and focus groups, topics related to accessibility, opportunities for career development, housing, availability of utilities, road accessibility and distance, access to long-term upgrading, sufficient staffing, and equipment and supplies arose most frequently. The nuances provided by NMTs on factors they consider in their employment choices, as well as the interrelationships between these factors and feasibility in Malawi, were used to cluster, prioritize and define job attributes and levels for the DCE questionnaire. Several examples of the key topics discussed are described below.

Participants reflected on the lack of housing options in rural areas and the impact of housing shortages on their decisions to both choose and remain in a rural job, in particular as their personal lives evolved alongside their careers. As one NMT student noted in a focus group discussion: *"Before they consider increasing the staff, increase the houses that they will live in. You cannot have a nurse living with the village head because there is no accommodation for her. In some areas there are not even houses that you can rent. A nurse cannot stay in a place like that [. . .] But if the health center has more good housing, electricity and passable roads then that will help when considering increasing the numbers [of health workers]."*

While salary was an important consideration, salary alone was not a sufficient motivating factor for respondents, in particular where housing, electricity and water were unavailable. As

one NMT shared: *"When people are aware that there are good houses, with potable water and electricity, they get motivated and rush to that place. But when there is a good salary, and the living conditions are pathetic, you will still think of your life first and decline to go to such a place. But good houses, potable water and electricity are a priority, and these are good motivating factors."* (Interviewee, male, rural MoH facility).

In addition to housing, issues of facility quality, supportive management and mentorship, and a sufficient size team to manage workload also emerged repeatedly throughout the interviews and FGDs. Respondents discussed the importance of these factors, not only for their own motivation to stay in a rural health facility, but also for the quality of care they were able to provide to patients. For example, one NMT shared: *"The other thing is that teamwork boosts the quality of care provided to the patients because where you don't know, a colleague will show you what to do. But in the rural [area] who will you ask? This is very important to consider when working."* (Interviewee, female, rural MoH facility). Another NMT highlighted: *"It makes me sad that we only wish that we had some of the equipment within the rural health facility to meaningfully save lives. So many times, I have seen cases where the referral system has failed and patients have died when such deaths could have been avoided if nurses in the rural area were equipped to a minimum. This to me, as a nurse and a human being, has an effect, that I would have saved a life but couldn't because of limitations. You cannot live everyday with such kinds of regrets. It is very sad and therefore you sometimes decide to move and go where you are able to manage to give the best you can."* (Interviewee, female, urban CHAM facility).

### DCE sample characteristics

For the DCE questionnaire, data was collected from 472 respondents, including 179 (37.9%) practicing NMTs and 293 (62.1%) NMT students. The participants were 65% female, 67.2% were 29 years or younger, and 86.8% had lived in a rural area. These proportions are broadly reflective of characteristics of Malawi's NMT workforce. Table 2 presents the demographic characteristics.

### Likelihood of working in a rural area

As shown in Table 2, 70.2% of participants reported they were either "very likely" or "likely" to work in a rural area in the future. Through bivariate logistic regression analysis we explored associations between stated likelihood of working in a rural area, and demographic characteristics and rural work experiences (see Table 3). There were significant associations between self-reported likelihood of working in a rural area and rural living experience, bonding agreements, and positive experiences working in rural areas previously. Students had two times higher odds of reporting they were "likely" or "very likely" to work in a rural area compared with currently practicing NMTs (odds ratio [OR] 2.20, confidence interval [CI] 1.35–3.59, p<0.01). NMTs and NMT students who had lived in a rural area previously (OR 3.40, CI 1.71–6.79, p<0.01), and those who had a bonding agreement with the government which states terms for compulsory service after graduation (OR 3.66, CI 2.22–6.04, p<0.01) were significantly more likely to report they were "likely" or "very likely" to work in a rural area in the future than their comparators. Those with "very good" experience working in a rural area previously were 6.14 times as likely (CI 1.89–19.93, p<0.01), and those with "good" experience were 7.50 times as likely (CI 2.33–24.09, p<0.01) to report they were "likely" or "very likely" to work in a rural area in the future as compared to those with a poor prior experience working in a rural area.

### Impact of attributes on rural and urban job choice

The results of the conditional logit model showed that salary, housing and facility quality had the greatest impact on likelihood of choosing a rural job. As shown in Table 4, respondents

**Table 2. Participant demographic characteristics.**

| Characteristic | n (%)[1] |
|---|---|
| **Gender** | |
| Female | 307 (65.0) |
| Male | 165 (35.0) |
| **Age** | |
| 29 years or younger | 317 (67.2) |
| 30 to 39 years | 124 (26.3) |
| 40 years or older | 30 (6.4) |
| **Marital status** | |
| Not married | 235 (49.8) |
| Married | 171 (36.2) |
| **Dependents** | |
| Has dependents | 189 (40.1) |
| No dependents | 282 (59.9) |
| **Health worker status** | |
| Practicing NMT | 179 (37.9) |
| NMT student | 292 (61.9) |
| **Under bonding agreement** | |
| No | 211 (45.6) |
| Yes | 252 (54.4) |
| **Ever lived in rural area** | |
| No | 62 (13.2) |
| Yes | 409 (86.8) |
| **Experience working in rural area for more than 3 months** | |
| No | 270 (57.6) |
| Yes | 199 (42.4) |
| **Stated likelihood of working in a rural area in the future** | |
| Very likely | 79 (23.8) |
| Likely | 154 (46.4) |
| Unlikely | 64 (19.3) |
| Very unlikely | 35 (10.5) |

1. Percentages may not sum to 100% due to missing data and rounding.

were 6.67 times as likely to choose a rural job with a 50% salary increase compared to a job with the base salary (CI 5.66–8.08, $p < 0.01$). A 25% salary increase had a lesser impact on rural job choice (OR 1.78, CI 1.51–2.10, $p < 0.01$). Following a 50% salary increase, the second most impactful job attribute was superior housing, with respondents 2.04 times as likely to choose a rural job if superior housing was provided compared to no housing (CI 1.71–2.44, $p < 0.01$). Respondents were 1.70 times as likely to choose a rural job where there was advanced facility quality (CI 1.47–1.96, $p < 0.01$).

A 50% salary increase and improved facility quality were also the attributes with the greatest influence on urban job choice, albeit in different magnitudes than in the rural job scenarios. Respondents were 3.84 times as likely to choose an urban job with a 50% salary increase (CI 3.22–4.59, $p < 0.01$), and 1.98 times as likely with advanced facility quality (CI 1.69–2.33, $p < 0.01$). A 25% salary increase had similar impact on job preference in urban settings (OR 1.55, CI 1.29–1.85, $p < 0.01$). Following salary, housing, and facility quality, supportive management had a similar effect on job choice in both rural (OR 1.51, CI 1.32–1.73, $p < 0.01$) and urban (OR 1.52, CI 1.32–1.74, $p < 0.01$) scenarios.

**Table 3. Association between likelihood of working in a rural area and participant characteristics.**

| Characteristic | Likely or very likely to work in rural area n (%)[1] | Unlikely or very unlikely to work in rural area n (%)[1] | Odds Ratio (Confidence Interval) | p value |
|---|---|---|---|---|
| **Gender** | | | | |
| Male | 97 (76.4) | 30 (23.6) | 1.64 (0.99–2.71) | 0.05 |
| Female | 136 (66.3) | 69 (33.7) | ref. | |
| **Age** | | | | |
| 30 years or older | 92 (73.6) | 33 (26.4) | 1.30 (0.80–2.14) | 0.29 |
| 29 years or younger | 141 (68.1) | 66 (31.9) | ref. | |
| **Marital status** | | | | |
| Not married | 112 (72.3) | 43 (27.7) | 1.65 (0.81–2.19) | 0.02 |
| Married | 92 (66.2) | 47 (33.8) | ref. | |
| **Dependents** | | | | |
| Has children | 109 (70.3) | 46 (29.7) | 1.02 (0.64–1.64) | 0.93 |
| No children | 123 (69.9) | 53 (30.1) | ref. | |
| **Health worker status** | | | | |
| NMT student | 122 (78.7) | 33 (21.3) | 2.20 (1.35–3.59) | <0.01 |
| Practicing NMT | 111 (62.7) | 66 (37.3) | ref. | |
| **Ever lived in rural area** | | | | |
| Yes | 215 (73.4) | 78 (26.6) | 3.40 (1.71–6.79) | <0.01 |
| No | 17 (44.7) | 21 (55.3) | ref. | |
| **Worked in rural area for more than 3 months** | | | | |
| No | 119 (72.6) | 45 (27.4) | 1.26 (0.79–2.03) | 0.33 |
| Yes | 113 (67.7) | 54 (32.3) | ref. | |
| **Bonding agreement or other rural obligation** | | | | |
| Has bonding agreement or other obligation | 147 (82.1) | 32 (17.9) | 3.66 (2.22–6.04) | <0.01 |
| No bonding agreement or other obligation | 84 (55.6) | 67 (44.4) | ref. | |
| **Rating of experience working in rural area** | | | | |
| Excellent | 15 (65.2) | 8 (34.8) | 2.56 (0.80–8.14) | 0.11 |
| Very good | 27 (81.8) | 6 (18.2) | 6.14 (1.89–19.93) | <0.01 |
| Good | 33 (84.6) | 6 (15.4) | 7.50 (2.33–24.09) | <0.01 |
| Fair | 26 (57.8) | 19 (42.2) | 1.87 (0.70–4.96) | 0.21 |
| Poor | 11 (42.3) | 15 (57.7) | ref. | |

1. Percentages may not sum to 100% due to missing data and rounding.

The conditional logit model was also run separately for NMT students and practicing NMTs, as well as for the various demographic sub-groups of the population (results not shown). However, there were no significant differences in the impact of attributes on job preferences in these different analyses.

## Predicted job uptake

We used the coefficients from the conditional logit model to transform data into percentages of health workers estimated to take a rural job compared to an urban job with various incentives provided, presented in Fig 1. With all interventions set to the base level, 43.9% of

**Table 4. Determinants of job preferences.**

| Incentive category and level | Odds Ratio | 95% CI | p value |
|---|---|---|---|
| **Location** (reference = rural) | | | |
| Urban | 1.30 | 0.99–1.65 | 0.06 |
| *Rural Job Characteristics* | | | |
| **Salary** (reference = base salary only) | | | |
| Base salary + 25% top-up | 1.78 | 1.51–2.10 | <0.01 |
| Base salary + 50% top-up | 6.76 | 5.66–8.08 | <0.01 |
| **Housing** (reference = no housing) | | | |
| Basic housing provided | 1.54 | 1.29–1.83 | <0.01 |
| Superior housing provided | 2.04 | 1.71–2.44 | <0.01 |
| **Facility quality** (reference = basic) | | | |
| Advanced facility quality | 1.70 | 1.47–1.96 | <0.01 |
| **Access to education** (reference = after 4 years) | | | |
| After 3 years | 1.29 | 1.12–1.49 | <0.01 |
| **Workload** (reference = heavy) | | | |
| Manageable workload | 1.32 | 1.13–1.54 | <0.01 |
| **Management** (reference = not supportive) | | | |
| Supportive | 1.51 | 1.32–1.73 | <0.01 |
| **Choice of location** (reference = random) | | | |
| Choice of location | 1.19 | 1.03–1.37 | 0.02 |
| *Urban Job Characteristics* | | | |
| **Salary** (reference = base salary only) | | | |
| Base salary + 25% top-up | 1.55 | 1.29–1.85 | <0.01 |
| Base salary + 50% top-up | 3.84 | 3.22–4.59 | <0.01 |
| **Facility quality** (reference = basic) | | | |
| Advanced facility quality | 1.98 | 1.69–2.33 | <0.01 |
| **Access to education** (reference = after 4 years) | | | |
| After 3 years | 1.20 | 1.04–1.40 | 0.01 |
| **Workload** (reference = heavy) | | | |
| Manageable workload | 1.40 | 1.21–1.62 | <0.01 |
| **Management** (reference = not supportive) | | | |
| Supportive | 1.52 | 1.32–1.74 | <0.01 |
| **Choice of location** (reference = random) | | | |
| Choice of location | 1.20 | 1.04–1.40 | 0.01 |
| **Model diagnostics** | | | |
| Number of participants | 472 | | |
| Number of observations | 9402 | | |
| Log likelihood | -2702.40 | | |
| Pseudo $R^2$ | 0.1691 | | |
| AIC | 5448.80 | | |
| BIC | 5570.33 | | |
| Prob > chi$^2$ | <0.001 | | |

AIC, Akaike information criterion; BIC, Bayesian information criterion.

respondents would prefer the rural job and 56.1% the urban job. This increased to 61.5% of respondents who would prefer a rural job with free superior housing compared to an urban job with no housing. We created a composite attribute that included all facility-level

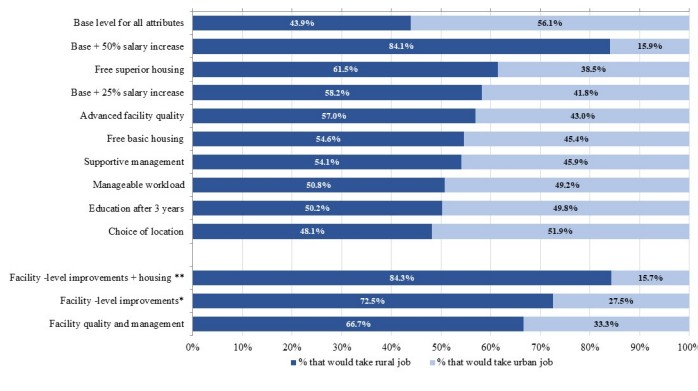

*Facility level improvements include advanced facility quality, manageable workload and supportive management.

** Includes all facility-level improvements plus free superior housing.

**Fig 1. Expected rural and urban job uptake with job characteristics.**

improvements (advanced facility quality, manageable workload, and supportive management). With all facility level improvements set to the highest level, 72.5% of respondents would be expected to select a rural job compared to an urban job without these improvements. With only improved facility quality and supportive management, and excluding hiring additional health workers to improve workload, 66.7% of respondents would be expected to select the rural job.

## Cost of incentives

We calculated the total cost of implementing each intervention per working year of an individual health worker using details of the attribute descriptions to generate assumptions on cost items, and validating assumptions and unit costs with the MoH. By combining the DCE results on increased odds of selecting a particular job with incentive cost per year, we estimated the additional or marginal cost of implementing each incentive compared to the cost of the base level and applied these marginal costs to the rural job uptake percentages to calculate a marginal cost per percentage point increase in rural job uptake.

The annual salary for an NMT was used, based on the most recent salary bands from 2014 with top-ups calculated from the gross total. We used average construction costs based on MoH and CHAI experience constructing staff housing for basic and superior public sector housing, assuming occupancy for 25 years and maintenance costs at 30% of total construction costs spread over 25 years. Upgrading costs assume payment of salary while the health worker is on study leave for two years, divided by the number of years the individual worked prior to school leave. Supportive management is assumed to be achieved through increased funding for district-level supervision and mentorship, costed as procurement and maintenance of one vehicle per district and costs associated with monthly supervision visits from district-level managers. Improved workload would be achieved by increasing the number of NMTs per site from the current average of two for a health center to three, and is costed as the annual cost of adding an additional NMT to a site. Facility quality and choice of job location were not costed due to insufficient standard assumptions to cost these interventions.

As shown in Table 5, the lowest total cost interventions per health worker per year were those related to facility-level improvements, including supportive management ($332) and manageable workload ($451). Whereas the highest costs per health worker per year were related to individual-level benefits of salary increase ($2,624 for 25% increase, $3,138 for 50% increase), free superior housing ($1,282), and educational upgrading opportunities after three

**Table 5. Cost per percentage point increase in rural job uptake.**

| Job attribute | Total cost per health worker per year | Marginal cost (compared to base) | % that would take rural job | Percentage point increase | Marginal cost per percentage point increase |
|---|---|---|---|---|---|
| **Salary** | | | | | |
| Base salary | $2,110 | - | - | - | - |
| Base salary + 25% top-up | $2,624 | $514 | 58.2% | 14.3 | $36 |
| Base salary + 50% top-up | $3,138 | $1,028 | 84.1% | 40.2 | $26 |
| **Housing** | | | | | |
| No housing provided | - | - | - | - | - |
| Free basic housing provided | $855 | $855 | 54.6% | 10.7 | $80 |
| Free superior housing provided | $1,282 | $1,282 | 61.5% | 17.6 | $73 |
| **Facility quality** | | | | | |
| Basic | *Not costed* | | - | - | - |
| Advanced | *Not costed* | | 57.0% | 13.1 | - |
| **Access to long-term upgrading opportunities** | | | | | |
| Eligible to apply after 4 years | $676 | - | - | - | - |
| Eligible to apply after 3 years | $902 | $352 | 50.2% | 6.3 | $56 |
| **Supportive management** | | | | | |
| Unsupportive management | - | - | - | - | - |
| Supportive management | $332 | $332 | 54.1% | 10.2 | $33 |
| **Choice of location** | | | | | |
| Randomly assigned to health facility | *Not costed* | | - | - | - |
| Choice of district | *Not costed* | | 48.1% | 4.2 | - |
| **Workload** | | | | | |
| Heavy workload | - | - | - | - | - |
| Manageable workload | $451 | $703 | 50.8% | 6.9 | $102 |

years ($902). However, the marginal costs of a 25% salary increase compared to the base ($514) and eligibility for upgrading after three years ($352) are among the lowest marginal costs. Salary top-ups and supportive management have the lowest cost per percentage point increase in job uptake. A salary increase of 25% is estimated to lead to a 14.3 percentage point increase in rural job uptake and a marginal cost per percentage point increase of $36, while improved supportive management is estimated to lead to a 10.2 percentage point increase in rural job uptake with a marginal cost per percentage point of $33. Our analysis predicts a 17.6 percentage point increase in rural job uptake with provision of superior housing at a cost of $73 per percentage point increase.

## Discussion

The results of this study indicate that salary, housing, and facility quality interventions had the greatest impact on rural job choice and retention. Salary and improved facility quality were also the attributes with greatest influence on urban job choice and retention, though in different magnitudes than the rural scenario. These findings are consistent with past qualitative assessments in Malawi, but offer more nuanced information on probability of choice, expected uptake, and cost of interventions. As the Government of Malawi considers development of a national retention strategy in line with recommendations in its HRH Strategic Plan 2017–

2022, these findings provide evidence to inform national policy design shaped by health worker preferences and feasibility of interventions.

While a 50% salary increase exerted the greatest influence on job choice, it is unlikely to be implemented in Malawi's context, where a large portion of the public sector health budget is spent on salaries and the allocation is unlikely to increase significantly [25]. Wages bills generally absorb a large proportion of total spending in LMICs, and therefore increases in compensation can have adverse consequences on the fiscal balance [26]. Malawi, as with many LMICs, pays health workers on the civil service scale, and salaries must be carefully managed across sectors to contain overall government spending [27]. Recognizing these fiscal constraints, the 50% salary increase attribute was included to allow for more detailed cost comparisons in our analysis, though was not prioritized by government. A 25% salary increase, more feasible in Malawi, was less impactful than other non-monetary interventions in the DCE. In addition to limited cost-effectiveness [28], research has suggested that increased salaries alone may not be sufficient to address health worker motivation and retention, and that nonfinancial incentives can significantly influence health worker motivation [29].

This study found that health workers were 2.04 times as likely to select a rural job if superior housing was provided. As highlighted in the qualitative findings, in rural areas in Malawi there are limited options for health workers to find rental housing, making it impractical for health workers to remain in rural areas long-term unless housing is provided. The 2016 MoH infrastructure assessment revealed a critical shortage of staff housing at nearly all health facilities. Recognizing this challenge, the Health Sector Capital Investment Plan 2017–2022 prioritizes construction of staff housing following the government's Umoyo Housing model [30]. The superior housing intervention included in our study is designed and costed following the Umoyo Housing specifications, a detached house with electricity and three bedrooms. While housing is a high priority and impactful long-term investment, it will require significant investment by government and development partners. To reduce overall costs and encourage community ownership, government and development partners can consider a community engagement approach in housing construction, such as including local communities as part of the labor force and in planning and governance of housing projects, and mobilizing local building materials. This approach has been successfully utilized in the education sector to build teacher housing in rural areas.

Facility quality, defined in our study as reliable infrastructure and available essential supplies, remains a significant barrier to service delivery in Malawi. In 2015/16, an average of only 24% of facilities were able to maintain enough stocks to cover 1 to 3 months for 23 tracer medicines and supplies, and only 63% had regular electricity [31]. Our findings highlight that improvements to facility quality are impactful in rural job choice, mirroring qualitative findings where NMTs emphasized that poor facility quality limits their ability to effectively treat patients, which significantly impacts their willingness to choose and remain in rural jobs. Combined interventions that mix several incentives can be highly motivating to health workers in their job choice, retention, and performance [32]. We designed a composite facility-level intervention, including supportive management, facility quality, and manageable workload attributes, which had a high impact on job uptake with 72.5% of participants expected to take a rural job with these conditions compared to an urban job without these improvements. Facility quality interventions can have far reaching impact on health worker motivation and retention, while also improving health worker performance and patient health outcomes [33]. This aligns with MoH priorities, where the government has recently established a Quality Management Directorate to provide leadership and coordinate quality management and improvement initiatives across the health sector, including a focus on facility-level quality improvement.

Though opportunities for career upgrading were discussed in the qualitative interviews and FGDs, and are frequently highlighted in retention literature, upgrading was a less impactful

intervention according to the results of our DCE. This may be due to a number of factors. The time to upgrade was reduced by only one year in the incentive (3 compared to 4 years of service prior to upgrading), which many not have been seen as a significant reduction. Further, at the time of the study, an upgrading NMT would receive their Registered Nurse (RN) Diploma and reenter the workforce at the same level, and thus NMTs may not have seen this as a significant pathway for career growth. NMTs are now able to upgrade to Degree-level nurses, which offers significant career advancement, and it would therefore be interesting to re-examine the impact of upgrading on NMTs' career choices with this new pathway available.

## Limitations

There are several limitations to this study. While the attributes we included in the DCE are reflective of health worker preferences, as articulated during FGDs and interviews and aligned with national and regional literature, there may be other retention interventions that would be meaningful to health workers in Malawi which were not included in our study. We sought to select interventions that were highly ranked by health workers, while also deemed feasible and aligned with government priorities, to ensure findings were applicable to policy discussions. While a DCE aims to present plausible scenarios, as with any model, a DCE cannot capture all complexities of real-world choices. Further, motivation to choose and remain in a job are jointly considered in this study as they are influenced by interrelated factors [7], however, the decision to remain in a job is complex and may change overtime as a health worker gains more experience, and these nuances cannot be fully captured by a DCE.

This DCE specifically focused on the NMT cadre, based on the high priority given to this cadre within Malawi's overall workforce strategy. While the findings have broad relevance in policy development, in particular for mid-level, rural providers, health workers in different cadres may have different preferences. In addition, while costing adds a unique dimension to the study, facility quality and choice of job location could not be costed due to insufficient standard assumptions and therefore are excluded from cost comparisons.

Finally, 79 people (16.7%) expressed a dominant preference for a certain job characteristic, 62 for rural jobs, and 17 for urban jobs. A dominant preference occurs when a respondent always selects a job based on one attribute irrespective of other attributes, and thus is unlikely to be influenced by any attribute. We excluded these individuals from our analysis as their responses violate the model assumptions and therefore cannot be used in the modeling approaches used for DCE analysis [22].

## Conclusion

The MoH has tested retention interventions over the past two decades and is committed to developing a national health worker retention policy to address Malawi's critical workforce shortages and inequitable distribution of health workers. Our study builds on previous national and regional literature which highlights factors that are important to mid-level providers in job choice and retention, and adds quantitative data on probability of choice, predicted uptake, and cost. Our study considers a range of monetary and non-monetary incentives that are feasible from a policy perspective and have the potential to influence health worker job choice, retention, motivation and performance. Our results indicate that housing and interrelated facility-level improvements would have the greatest impact on rural job choice, while also creating longer-term improvements to health workers' working and living environment. These results provide practical evidence for policymakers in Malawi to use in the design of national retention strategies, and can be used beyond Malawi to support policy discussions on workforce recruitment and retention.

## Supporting information

**S1 File. Qualitative interview and focus group discussion guides.**
(DOCX)

**S2 File. Twelve choice sets in Block A of the DCE questionnaire.**
(DOCX)

**S3 File. DCE participant information and choice selections.** This file contains one record for each participant in the study including demographic and survey data. This dataset shows whether the participant received DCE Block A or B, and whether they chose job option 1 or 2 in the 12 job choice sets provided. This must be combined with S4 File for full DCE analysis.
(DTA)

**S4 File. Reference data file for DCE job sets.** This file contains reference information about the choice sets provided to participants in the DCE questionnaire. There were 2 blocks of 12 job choice sets, with each job choice set containing 2 alternatives. This file therefore contains 48 records with the details for each job alternative offered. For each of the incentive categories (housing, facility quality, upgrading, workload and management) each job alternative included one incentive level, which is denoted with the value 1.
(DTA)

## Acknowledgments

The authors would like to acknowledge the MoH, CHAM, and NMT training colleges for their contribution to the success of this study. We would also like to acknowledge the interview, FGD and DCE questionnaire participants from the participating schools and health facilities for sharing their valuable insights. We would like to thank Lauren Reising for her review of the manuscript. Finally, we would like to express appreciation to Peter Rockers (Boston University) and Duane Blaauw (Wits University) for guidance and technical support in the analysis process and writing of the original technical report.

## Author Contributions

**Conceptualization:** Levison Nkhoma, Courtney McKay, Dalitso Kabambe, Andrews Gunda.

**Data curation:** Leslie Berman, Levison Nkhoma, Courtney McKay.

**Formal analysis:** Levison Nkhoma, Margaret Prust, Courtney McKay.

**Methodology:** Margaret Prust, Courtney McKay.

**Project administration:** Leslie Berman, Levison Nkhoma, Margaret Prust, Mihereteab Teshome.

**Supervision:** Leslie Berman, Levison Nkhoma, Margaret Prust, Mihereteab Teshome, Dumisani Banda, Dalitso Kabambe, Andrews Gunda.

**Validation:** Leslie Berman.

**Writing – original draft:** Leslie Berman.

**Writing – review & editing:** Leslie Berman, Levison Nkhoma, Margaret Prust, Courtney McKay, Mihereteab Teshome, Dumisani Banda, Dalitso Kabambe, Andrews Gunda.

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
