## [Decision Letter · Decision Letter 0]

18 Jan 2021

PONE-D-20-33356

Analysis of policy interventions to attract and retain nurse midwives in rural areas of Malawi: A discrete choice experiment

PLOS ONE

Dear Author,

Thank you for submitting your manuscript to PLOS ONE. After careful consideration, we feel that it has merit but does not fully meet PLOS ONE’s publication criteria as it currently stands. Therefore, we invite you to submit a revised version of the manuscript that addresses the points raised during the review process.

We look forward to receiving your revised manuscript.

Kind regards,

Ramesh Kumar, PhD

Academic Editor

PLOS ONE

Journal Requirements:

2. When reporting the results of qualitative research, we suggest consulting the COREQ guidelines: http://intqhc.oxfordjournals.org/content/19/6/349. In this case, please consider including more information on the number of interviewers, their training and characteristics; and please provide the interview guide used.

Furthermore, in your Methods section, please provide a justification for the sample size used in your study, including any relevant power calculations (if applicable)

3.Thank you for stating the following in the Financial Disclosure section:

"This study was made possible with financial support from the Norwegian Ministry of Foreign Affairs through a grant from the Royal Norwegian Embassy of Malawi (https://www.norway.no/en/malawi/) to the Clinton Health Access Initiative Malawi. The funders had no role in study design, data collection and analysis, decision to publish, or preparation of the manuscript."

We note that one or more of the authors are employed by a commercial company: Clinton Health Access Initiative, Inc. and Analytics and Implementation Research Team, Clinton Health Access Initiative, Inc.

Additional Editor Comments:

Please see comments and revise your paper accordingly.

Reviewers' comments:

Reviewer's Responses to Questions

**Comments to the Author**

1. Is the manuscript technically sound, and do the data support the conclusions?

Reviewer #1: Partly

Reviewer #2: Yes

2. Has the statistical analysis been performed appropriately and rigorously? 

Reviewer #1: Yes

Reviewer #2: Yes

3. Have the authors made all data underlying the findings in their manuscript fully available?

Reviewer #1: Yes

Reviewer #2: Yes

4. Is the manuscript presented in an intelligible fashion and written in standard English?

Reviewer #1: Yes

Reviewer #2: Yes

5. Review Comments to the Author

Reviewer #1: Dear Corresponding Author, Please include comments highlighted

Plos one: Analysis of policy interventions to attract and retain nurse midwives in rural areas of Malawi: A discrete choice experiment

Manuscript Number: PONE-D-20-33356

Corresponding Author: Leslie Renee Berman Clinton Health Access Initiative Lilongwe 3, MALAWI

Comments:

1. Line 23 Background not clearly indicated problem of study in abstract section

2. Line 34: In methodology section conditional logit regression model assumption not well described. Include comments accordingly

3. Line 45 – Include qualitative findings in the result section of the abstract

4. Line 53- Indicate gaps in the introduction section

5. Line 101 Study area not described in the method section. In addition level of retention of midwives from reports in the study area not stated. Elaborate this issues accordingly.

6. Line 118: How FGD and key informant participants selected in the study?

7. Line 177: How you calculated 472 samples in this study? What are assumptions used.

8. Line194: What is likelihood of working in rural areas? How you measured it? It needs clarification

9. Line 240: ‘House is motivating factors’. Do you think this is right? Explain it

10. General Questions? How you engaged stakeholders during policy analysis? What type of policy analysis methodology used? Forward your answer accordingly

11. Line 279: Why research team members not included confidence interval while reporting significant variables

12. Line 320: Give Operational definitions for quality …

Reviewer #2: Comments on PONE-D-20-33356 “Analysis of policy interventions to attract and retain nurse midwives in rural areas of Malawi: A discrete choice experiment”

This paper uses responses from 472 nurse midwife technicians and students to assess the relative importance of various factors influencing the decision to accept employment in rural areas. Data were collected in Fall 2016. Results suggest that housing provision is the most important factor influencing the decision to locate to a rural area. The analysis is based on respondent selection between two hypothetical jobs, one urban and one rural, with each job having a random set of job attributes. Each respondent chose between 12 pairs of jobs. The analysis then examined how probability of rural job selection was influenced by the job attributes.

Comments

1) The paper argues (page 11) that it is a violation of random utility theory for a respondent to always choose the higher paying option as there must be tradeoffs between pay and job attributes. That is not true. If the respondent does not care about the other job attributes, they will always select the higher paying option. The analysis assumes that the nonpecuniary job factors raise worker utility, but that is not necessarily true.

2) The paper uses acronyms to excess. OR for odds ratios, FGD for focs group discussion, and so on. This gets annoying. You should use Odds Ratio in the tables and not the acronym

3) Table 3 should be a multinomial logit model, I think, but no goodness of fit statistics are presented. If it is just a series of separate bivariate relationships, it is not very useful.

4) It is not clear that the demographic variables included in Table 3 are also included in Table 4. If the model is set up as a multinomial logit, the demographic variables can be included. As a conditional logit, it may be that only the job attributes are allowed to affect the choice unless the demographic variables are interacted with the job attributes. It seems that the importance of housing or salary to the choice of rural area would differ by gender, marital status, and presence of children

5) If the authors want to stick with the conditional logit, they could replicate the tradeoffs shown in table 5 by marital status, gender, and rural upbringing. I suspect the results would be quite interesting. Urban origin would require greater payoffs to accept a rural posting, for example.

6. PLOS authors have the option to publish the peer review history of their article (what does this mean?). If published, this will include your full peer review and any attached files.

Reviewer #1: **Yes: **Adem Abdulkadir Abdi

Reviewer #2: No

---

## [Author Response · Author response to Decision Letter 0]

14 Feb 2021

Below please find a response to each point raised by the academic editor and peer reviewers. The line numbers referenced in our responses below correspond to edits we have made in the “Revised Manuscript with Track Changes” in response to the reviewer comments. 

Points Raised by Academic Editor 

Response: We have reviewed the PLOS ONE style requirements and adjusted the manuscript accordingly, including renaming files as per the requirements for file naming. 

2. When reporting the results of qualitative research, we suggest consulting the COREQ guidelines: http://intqhc.oxfordjournals.org/content/19/6/349. In this case, please consider including more information on the number of interviewers, their training and characteristics; and please provide the interview guide used. Furthermore, in your Methods section, please provide a justification for the sample size used in your study, including any relevant power calculations (if applicable). 

Response: We have reviewed the COREQ guidelines as suggested, and for the qualitative research we have added information on the number and background of interviewers in the Methods section (line 145), and have uploaded the interview and focus group discussion guides as Supporting Information File 1 (S1 File). We have also added additional information on the rationale for the qualitative sample size (line 128), and the sample size calculation for the DCE questionnaire (line 195). More detail on the sample size calculation for the DCE questionnaire is provided below in this document in response to Reviewer #1, Question 7.

3. We note that one or more of the authors are employed by a commercial company: Clinton Health Access Initiative, Inc. and Analytics and Implementation Research Team, Clinton Health Access Initiative, Inc. 

a. Please provide an amended Funding Statement declaring this commercial affiliation, as well as a statement regarding the Role of Funders in your study.

b. Please also provide an updated Competing Interests Statement declaring this commercial affiliation along with any other relevant declarations relating to employment, consultancy, patents, products in development, or marketed products, etc. Within your Competing Interests Statement, please confirm that this commercial affiliation does not alter your adherence to all PLOS ONE policies on sharing data and materials.

Response: We have included an updated Funding Statement and Competing Interests Statement in our cover letter. In the Funding Statement we have indicated that six authors have an affiliation as employees of the Clinton Health Access Initiative, and added a statement on the role of these authors in the design and implementation of the study and publication of this manuscript. We have also updated our Competing Interests Statement to declare this professional affiliation. In the Competing Interests Statement, we have also confirmed that this affiliation does not alter our adherence to PLOS ONE policies on sharing data and materials.

4. While revising your submission, please upload your figure files to the Preflight Analysis and Conversion Engine (PACE) digital diagnostic tool, https://pacev2.apexcovantage.com/.

Response: We have uploaded the figure (Fig 1) to the PACE digital diagnostic tool, and have attached the PACE corrected image along with our submission. 

Points Raised by Reviewer #1

1. Line 23: Background not clearly indicated problem of study in abstract section. 

Response: We have added a sentence in the background section of the abstract which further elaborates the problem statement (line 25). 

2. Line 34: In methodology section conditional logit regression model assumption not well described. Include comments accordingly. 

Response: We have added a sentence in the methods section of the abstract which provides more detail on the conditional logit regression model (line 38). 

3. Line 45: Include qualitative findings in the result section of the abstract. 

Response: We have added a sentence in the results section of the abstract which summarizes the qualitative findings (line 45). 

4. Line 53: Indicate gaps in the introduction section. 

Response: In the introduction section of the manuscript we have aimed to highlight gaps in the health workforce environment in Malawi, including high vacancy rates, unequal distribution of health workers between urban and rural areas, and high rates of attrition. We have also noted gaps in the existing literature which this study aims to fill in order to provide evidence for policymakers to develop context-specific rural retention strategies. In particular, we explain that while previous evidence aimed to understand drivers of health worker retention in general, a mixed-methods investigation of interventions that can influence rural job uptake and retention and costs of interventions is needed, with a focus on mid-level providers who are the frontline workers in rural facilities in Malawi. 

5. Line 101: Study area not described in the method section. In addition, level of retention of midwives from reports in the study area not stated. Elaborate these issues accordingly.

Response: For the focus groups, 12 randomly selected NMT students were drawn from three purposively selected NMT training institutions, one in each of the country’s three regions (north, central, south). For interviews, one district from each region was purposively selected and within each of the three selected districts four facilities were chosen, including one with each of the following classifications: Ministry of Health (MoH) rural, MoH urban, Christian Health Association of Malawi (CHAM) rural and CHAM urban. Within those facilities, one NMT was selected who was available at the time of interview. Please see the paragraph which begins on line 127. 

For the DCE questionnaire, all graduating NMT students were invited to participate from five randomly selected training institutions of the 12 training institutions in-country, representing geographic diversity across the country’s five administrative zones (north, central west, central east, southwest, southeast). For practicing NMTs, ten districts were randomly selected, including two districts from each of the five zones. From each district, six rural facilities and one urban facility were randomly selected. At each health facility, all NMTs present on the day of the survey were invited to participate. Please see the paragraph which begins on line 205. 

Regarding attrition rates, it is difficult to calculate “level of retention” of nurse midwives as the Government of Malawi does not routinely collect these data, and it is not possible to disaggregate national retention estimates by cadre. However, estimates in Malawi’s national Human Resources for Health Strategic Plan indicate an attrition rate of between 3-15%, noting that this is likely an underestimate. We have included this information on attrition rates in the background section on line 82. 

6. Line 118: How FGD and key informant participants selected in the study?

Response: The sampling methods for interview and focus group participants are explained in the methods sub-section titled “Qualitative methods and questionnaire development”. Regarding focus group participants, at the time of the study there were 12 schools that trained NMTs in Malawi, and one school was purposively sampled from each of the country’s three regions (north, central, south), and 12 graduating students from each school were invited to the focus groups. The 12 students were randomly selected from the class register. For interviews, one district from each region was selected, and four facilities were chosen per district that met the classifications MoH Rural, MoH Urban, CHAM rural and CHAM Urban. One NMT was selected per facility, depending on their availability for the scheduled day of the interview. 

7. Line 177: How you calculated 472 samples in this study? What are assumptions used.

Response: We have added an explanation for how the sample size was calculated for the DCE questionnaire beginning on line 195. Literature indicates that obtaining reliable estimates of preferences requires at least 30 to 50 respondents per sub-group to be analyzed. The appropriate sample size is impacted by the number of attributes to be analyzed, the number of options within each attribute, and the number of question sets provided to each participant. Other DCE sample size analyses have concluded that the overall precision of DCE parameters is impacted by a balance between statistical efficiency, which minimizes the confidence interval around a parameter estimate for a given sample size, and response efficiency, which minimizes measurement errors resulting from respondents’ inattention to choice questions. While large sample sizes yield smaller confidence intervals around parameters, caution needs to be exercised in resource constrained settings where such samples sizes may not be feasible. Using multiple DCEs with simulated sample sizes to estimate the sample size where precision would improve, a DCE methodological review revealed that precision steadily increases when the sample size is below 150 and flattens at around 300 observations and above. We therefore aimed to sample a minimum of 300 participants, by recruiting all graduating NMTs from five randomly selected training institutions, and all available NMTs at seven randomly selected health facilities (six rural, one urban) in each of ten randomly selected districts. 

8. Line 194: What is likelihood of working in rural areas? How you measured it? It needs clarification. 

Response: Likelihood of working in a rural area is a self-reported measure by respondents to the DCE questionnaire. Respondents were asked, “Please rate how likely you are to continue working in a rural facility or to start working at a rural facility at some point in the future” with Likert Scale response options from “Very Likely” to “Very Unlikely”. We have clarified this in the text on line 221. 

9. Line 240: ‘House is motivating factors’. Do you think this is right? Explain it.

Response: This statement is a direct quote from a respondent where they indicate, “But good houses, portable water and electricity are a priority, these are good motivating factors.” A central finding of the paper, both from the qualitative and quantitative data, is that provision of free superior housing, which includes water and electricity, is a motivating factor to attract and retain health workers in rural areas. We have discussed the implications of this finding throughout the discussion section. 

10. General Questions? How you engaged stakeholders during policy analysis? What type of policy analysis methodology used? Forward your answer accordingly. 

Response: We created a listing of all the coded job attributes that emerged from the qualitative data collection and grouped them into common thematic areas. We then asked government stakeholders to rank these attributes using several inclusion criteria such as feasibility of implementing the intervention in Malawi’s context and importance/strength of the preferences based on qualitative interviews. We have added more detail on this in the methods section where we discuss stakeholder consultations beginning on line 156. 

11. Line 279: Why research team members not included confidence interval while reporting significant variables? 

Response: We have included confidence intervals for all variables throughout the results section, in both the narrative and tables. 

12. Line 320: Give operational definitions for quality.

Response: The operational definition for “facility quality” is included in the methods section in Table 1 (line 168) where we provide definitions for all job attributes and attribute levels. Basic facility quality includes unreliable electricity, and equipment, drugs and supplies that are not always available. Advanced facility quality includes reliable electricity and equipment, drugs and supplies that are always available. 

Points Raised by Reviewer #2 

1. The paper argues (page 11) that it is a violation of random utility theory for a respondent to always choose the higher paying option as there must be tradeoffs between pay and job attributes. That is not true. If the respondent does not care about the other job attributes, they will always select the higher paying option. The analysis assumes that the nonpecuniary job factors raise worker utility, but that is not necessarily true. 

Response: We have clarified this statement from line 235. We agree that a respondent may have a preference for only one attribute, and therefore would always make choices based on this attribute. However, respondents who express a dominant preference and make selections based on only one attribute violate the model assumptions. As noted in the literature, a DCE model is not equipped to handle such responses. Therefore, we excluded the 79 respondents who expressed a dominant preference for a certain job characteristic from our analysis, as their responses violate the model assumptions. 

2. The paper uses acronyms to excess. OR for odds ratios, FGD for focus group discussion, and so on. This gets annoying. You should use Odds Ratio in the tables and not the acronym. 

Response: We have removed certain acronyms throughout the paper, including the use of FGD for focus group discussion. We have changed “OR” to “Odds Ratio” in the table headings. 

3. Table 3 should be a multinomial logit model, I think, but no goodness of fit statistics are presented. If it is just a series of separate bivariate relationships, it is not very useful.

Response: Table 3 shows the values for bivariate logistic regressions comparing each sub-group to show how self-stated likelihood of working in a rural area in the future varies by level of demographic characteristics. We have clarified this in the manuscript on line 300. As the focus of our research was on the responses to the DCE survey itself, we did not conduct extensive analysis on participants’ responses to introductory and demographic questions on the questionnaire, such as stated likelihood of working in rural areas. However, we do believe these bivariate relationships add value, and have therefore included them in the paper, as they indicate significant associations between certain demographic characteristics and stated likelihood of working in a rural area in the future, which may be relevant to policymakers and implementers aiming to design, and more appropriately target, retention interventions. 

4. It is not clear that the demographic variables included in Table 3 are also included in Table 4. If the model is set up as a multinomial logit, the demographic variables can be included. As a conditional logit, it may be that only the job attributes are allowed to affect the choice unless the demographic variables are interacted with the job attributes. It seems that the importance of housing or salary to the choice of rural area would differ by gender, marital status, and presence of children. 

Response: As points 4 and 5 are interrelated, we have responded jointly to these two points. Please see our response underneath question 5 below. 

5. If the authors want to stick with the conditional logit, they could replicate the tradeoffs shown in table 5 by marital status, gender, and rural upbringing. I suspect the results would be quite interesting. Urban origin would require greater payoffs to accept a rural posting, for example. 

Response: Through literature review and expert consultation, we decided to use a conditional logit model, as one of the most widely used analytical approaches for DCEs. To investigate the potential impact of demographic variables on job attributes, we ran separate conditional logit models for each of the various demographic sub-groups of the population. As we did not find meaningful differences in the results of these separate analyses, we have not presented these results in the manuscript. We have added this information to the methods section on line 235 and in the results section on line 340.

---

## [Decision Letter · Decision Letter 1]

9 Apr 2021

PONE-D-20-33356R1

Analysis of policy interventions to attract and retain nurse midwives in rural areas of Malawi: A discrete choice experiment

PLOS ONE

Dear Author,

Thank you for submitting your manuscript to PLOS ONE. After careful consideration, we feel that it has merit but does not fully meet PLOS ONE’s publication criteria as it currently stands. Therefore, we invite you to submit a revised version of the manuscript that addresses the points raised during the review process.

We look forward to receiving your revised manuscript.

Kind regards,

Ramesh Kumar, PhD

Academic Editor

PLOS ONE

Journal Requirements:

Additional Editor Comments (if provided):

Reviewers' comments:

Reviewer's Responses to Questions

**Comments to the Author**

1. If the authors have adequately addressed your comments raised in a previous round of review and you feel that this manuscript is now acceptable for publication, you may indicate that here to bypass the “Comments to the Author” section, enter your conflict of interest statement in the “Confidential to Editor” section, and submit your "Accept" recommendation.

Reviewer #1: All comments have been addressed

Reviewer #2: All comments have been addressed

2. Is the manuscript technically sound, and do the data support the conclusions?

Reviewer #1: (No Response)

Reviewer #2: (No Response)

3. Has the statistical analysis been performed appropriately and rigorously? 

Reviewer #1: Yes

Reviewer #2: (No Response)

4. Have the authors made all data underlying the findings in their manuscript fully available?

Reviewer #1: Yes

Reviewer #2: (No Response)

5. Is the manuscript presented in an intelligible fashion and written in standard English?

Reviewer #1: Yes

Reviewer #2: (No Response)

6. Review Comments to the Author

Reviewer #1: Comments are addressed in manuscript line by line. But I advise you to improve following comments:

1. Indicate qualitative data analysis in the method section. In the result section please indicate sex and age of key informant interview result while reporting.

2. Please indicate assumptions of conditional logit regression and model fittness test undergone. In the sampling section if you included all study participants please amend sample size calculation area and improve to census type of sampling strategy.

3. Indicate Policy analysis steps used and how you engaged stakeholders in the policy analysis.

Reviewer #2: (No Response)

7. PLOS authors have the option to publish the peer review history of their article (what does this mean?). If published, this will include your full peer review and any attached files.

Reviewer #1: **Yes: **Adem Abdulkadir Abdi

Reviewer #2: No

---

## [Author Response · Author response to Decision Letter 1]

19 May 2021

Thank you for your careful review of our manuscript titled “Analysis of policy interventions to attract and retain nurse midwives in rural areas of Malawi: A discrete choice experiment.” Below please find a response to each point raised by the peer reviewer. The line numbers referenced in our responses below correspond to edits we have made in the “Revised Manuscript with Track Changes” in response to the reviewer comments. 

Points Raised by Reviewer 1

1. Indicate qualitative data analysis in the method section. In the result section please indicate sex and age of key informant interview result while reporting.

Response: We have expanded the qualitative data analysis described in the Methods section starting from line 148. We have also added in sub-headings to clearly demarcate the qualitative and quantitative data analysis sections presented in the Methods. We have added the sex of key informant interviewees in the Results section; however, we did not retain age information for key informant interviews. Focus group discussions were anonymized during transcription, and therefore it is not possible to disaggregate sex and age for individual speakers in the focus groups. Of the four quotations included in the Results, one is from a focus group, and we have clarified this in the manuscript. 

2. Please indicate assumptions of conditional logit regression and model fittness test undergone. In the sampling section if you included all study participants please amend sample size calculation area and improve to census type of sampling strategy.

Response: We have added the three key assumptions of the model to the DCE data analysis section on line 230 as follows: “The conditional logit model is based on three assumptions: (1) independence of irrelevant alternatives; (2) error terms are independent and identically distributed across observations; and (3) no preference heterogeneity across respondents.” Goodness-of-fit criteria, including Akaike and Bayesian information criteria and pseudo R2, were used to assess model fit. These measures are included in the Results section in Table 4. Regarding the sampling strategy, on line 210 we have indicated that to reach our minimum target sample size, we used a census approach to sampling all NMT students and practicing NMTs in a subset of randomly selected NMT training colleges and health facilities. 

3. Indicate policy analysis steps used and how you engaged stakeholders in the policy analysis.

Response: As elaborated from line 156 onwards, government and other stakeholders were engaged in a workshop to review the qualitative data outputs and prioritize which coded attributes should be included in the DCE questionnaire from a policy perspective, using specific selection criteria. The goal was to ensure that all final attributes included in the DCE questionnaire were interventions that could be practically implemented by the Government of Malawi in the future. As noted in the Discussion section, the final results of this study were then used to inform recommendations in Malawi’s HRH Strategic Plan.

---

## [Decision Letter · Decision Letter 2]

8 Jun 2021

Analysis of policy interventions to attract and retain nurse midwives in rural areas of Malawi: A discrete choice experiment

PONE-D-20-33356R2

Dear Author,

We’re pleased to inform you that your manuscript has been judged scientifically suitable for publication and will be formally accepted for publication once it meets all outstanding technical requirements.

Kind regards,

Ramesh Kumar, PhD

Academic Editor

PLOS ONE

Additional Editor Comments (optional):

Reviewers' comments:

Reviewer's Responses to Questions

**Comments to the Author**

1. If the authors have adequately addressed your comments raised in a previous round of review and you feel that this manuscript is now acceptable for publication, you may indicate that here to bypass the “Comments to the Author” section, enter your conflict of interest statement in the “Confidential to Editor” section, and submit your "Accept" recommendation.

Reviewer #1: All comments have been addressed

Reviewer #2: All comments have been addressed

2. Is the manuscript technically sound, and do the data support the conclusions?

Reviewer #1: Yes

Reviewer #2: (No Response)

3. Has the statistical analysis been performed appropriately and rigorously? 

Reviewer #1: Yes

Reviewer #2: (No Response)

4. Have the authors made all data underlying the findings in their manuscript fully available?

Reviewer #1: Yes

Reviewer #2: (No Response)

5. Is the manuscript presented in an intelligible fashion and written in standard English?

Reviewer #1: Yes

Reviewer #2: (No Response)

6. Review Comments to the Author

Reviewer #1: Majority of Comments raised in the first comments were addressed. But still not clearly indicated type of type of policy analysis performed. Try to shorten the manuscript

Reviewer #2: (No Response)

7. PLOS authors have the option to publish the peer review history of their article (what does this mean?). If published, this will include your full peer review and any attached files.

Reviewer #1: No

Reviewer #2: No

---

## [Editor Report · Acceptance letter]

10 Jun 2021

PONE-D-20-33356R2 

Analysis of policy interventions to attract and retain nurse midwives in rural areas of Malawi: A discrete choice experiment 

Dear Dr. Berman:

I'm pleased to inform you that your manuscript has been deemed suitable for publication in PLOS ONE. Congratulations! Your manuscript is now with our production department. 

Kind regards, 

on behalf of

Dr. Ramesh Kumar 

Academic Editor

PLOS ONE